# The RMaP challenge of predicting RNA modifications by nanopore sequencing
Jannes Spangenberg [1], Stefan Mündnich [2], Anne Busch [3], Stefan Pastore[2,4], Anna Wierczeiko[4], Winfried Goettsch [1,5], Vincent Dietrich[4], Leszek P. Pryszcz [6], Sonia Cruciani[6], Eva Maria Novoa[6,7,8], Kandarp Joshi [9,10], Ranjan Perera[9,10], Salvatore Di Giorgio [11], Paola Arrubarrena[12,13], Irem Tellioglu [11,14], Chi-Lam Poon[15,16], Yuk Kei Wan[17], Jonathan Göke[17,18], Andreas Hildebrandt [3], Christoph Dieterich[19] ✉, Mark Helm[2] ✉, Manja Marz[1,5,20] ✉, Susanne Gerber [4,21] ✉ & Nicolo Alagna [4] ✉

The field of epitranscriptomics is undergoing a technology-driven revolution. During past decades, RNA modifications like N6-methyladenosine (m⁶A), pseudouridine (ψ), and 5-methylcytosine (m⁵C) became acknowledged for playing critical roles in cellular processes. Direct RNA sequencing by Oxford Nanopore Technologies (ONT) enabled the detection of modifications in native RNA, by detecting noncanonical RNA nucleosides properties in raw data. Consequently, the field's cutting edge has a heavy component in computer science, opening new avenues of cooperation across the community, as exchanging data is as impactful as exchanging samples. Therefore, we seize the occasion to bring scientists together within the RNA Modification and Processing (RMaP) challenge to advance solutions for RNA modification detection and discuss ideas, problems and approaches. We show several computational methods to detect the most researched mRNA modifications (m⁶A, ψ, and m⁵C). Results demonstrate that a low prediction error and a high prediction accuracy can be achieved on these modifications across different approaches and algorithms. The RMaP challenge marks a substantial step towards improving algorithms' comparability, reliability, and consistency in RNA modification prediction. It points out the deficits in this young field that need to be addressed in further challenges.

Chemical modifications of RNA occur in the post- and co-transcriptional phase. They can modulate and shape cellular processes at different stages, from gene transcription to cellular life cycle[1–6]. In this context, a new field of research called epitranscriptomics has emerged in recent years[7–9], which, analogous to epigenetics, focuses on studying and understanding how molecular properties outside typical sequence information can regulate gene expression. One of the best-studied RNA modifications is N6-methyladenosine (m⁶A), which is known to have several regulatory functions[10,11]. Conversely, misregulation of m⁶A has been related to numerous diseases[12–14]. Together with two further prominent cases of RNA modifications, pseudouridine (Ψ)[3,15] and 5-methylcytosine (m⁵C)[1,16], m⁶A forms a triad that is under particular scrutiny in mRNA, where modifications are complicated to detect.

Pseudouridine is known to be the most abundant RNA modification in cellular RNA[17]. This modification has previously been shown to regulate RNA structure or alter mRNA functions by modulating non-canonical base

pairing and decoding[15,18–20]. Similarly to Ψ, m⁵C was initially associated with the functionality and regulation of tRNA and rRNA[21–23]. However, it has been recently discovered that m⁵C also plays an essential role in mRNA functionality[24,25]. However, apart from their biological significance, the three mentioned modifications are only three of more than 170 different chemical RNA modifications identified during the recent decades[1,2,26,27]. This number illustrates the complexity of the epitranscriptomic landscape and highlights the need to develop methods for identifying, characterizing, and differentiating individual RNA modifications. In this direction, several methods have already been developed to explore the effect of modification on the transcriptome. Selected examples include MeRIP-Seq[28], m6ACE-Seq[29], Pseudo-seq[30], miCLIP[31], and GLORI[32], which combine next-generation sequencing (NGS) technology with chemical treatments or antibodies to detect and characterize transcriptome-wide RNA modifications in short-reads. While these methods represent significant advances, they rely on cDNA synthesis rather than direct RNA sequencing and consequently lose

A full list of affiliations appears at the end of the paper. ✉e-mail: christoph.dieterich@uni-heidelberg.de; mhelm@uni-mainz.de; manja@uni-jena.de; sugerber@uni-mainz.de; nalagna@uni-mainz.de

information during each processing step. Also, the ability of short-read sequencing technologies to accurately capture the diversity and adaptability of RNA modifications is clearly limited[2,33,34]

In parallel, a new technology developed by Oxford Nanopore Technology (ONT)[35] based on direct RNA Sequencing (DRS) opened a new way to analyze and identify RNA modification at single nucleotide resolution in long reads[36–42]. ONT allows for identifying possible modifications on nucleotides crossing the pore via slight alterations in the measured current, leading to sequence-to-signal mismatching. Building on these observations, several DRS-based modification detection methods were developed. These can be split into two categories: Comparative and de novo detection (Table 1). A few examples of comparative methods are nanoRMS[41], EpiNano[43], Magnipore[44], xPore[45], nanocompore[46], Yanocomp[47], ELIGOS[39], Tombo from the ONT company, DRUMMER[48], DiffErr[49], and JACUSA2[50] which either compare the raw signal characteristics with a negative control to detect RNA modifications or use error patterns. On the other side, the de novo methods like nanoRMS[41], nanoDoc[51], m6anet[52], Nanom6A[53], DENA[54], mAFiA[55], Penguin[56], CHEUI[57], MINES[58], nano-ID[59], NanoPsu[60], NanoSPA[61], TandemMod[62], IL-AD[63] and ModiDeC[64] focus on training personalized deep neural networks using synthetic and labeled dataset from in vitro synthetic sequences or in vivo transcribed RNAs to obtain ground-truth labels for modifications. These typologies of methods allow us to successfully detect a specific RNA modification at a single base resolution. Despite the efficiency of all methods mentioned above, most are performed and tested on in loco-generated sequences, which can bring discontinuity in results when more than one method is used to evaluate a new dataset.

Focusing on this aspect, we present the RMaP challenge, where RNA modification methods can be jointly tested, evaluated, and compared using selected metrics with the purpose of evaluating both the performances of the submitted method and extracting key steps in the pipeline that can help in method development. In the RMaP challenge, we created specific synthetic datasets intending to focus on detecting and analyzing m6A, m5C, and Ψ RNA modifications. For each of these modifications, a unique data set combines designed sequences, in vitro transcription technique (IVT), and ONT technology to generate modified and unmodified DRS reads for comparison. A challenge was posed for each modification, in which the participants had to calculate the modification target frequencies (brief frequency, which is defined by dividing the number of modified bases by the number of total readings analyzed) at the level of the single nucleotide base in a specific time window of about 40 days. Like in a competition, data sets and tasks for the RMaP challenges were revealed on the starting day, and submissions were accepted only within the deadline to offer equal conditions and opportunities to all the participants. In Challenge 1, participants were given an m5C dataset along with a designed reference sequence, but no additional information was provided. In Challenge 2, an m6A dataset was provided, accompanied by the designed reference sequence. For both challenges, the goal was to predict target frequencies at each position in the reference sequence. Challenge 3 involved a Ψ dataset, which was split into two parts: one for training and the other for testing. The objective was to create and train a machine-learning algorithm to predict the modification. Predictions of target frequencies were then made on the test dataset. The prepared DRS datasets served as a common ground truth in the RMaP challenge, where the different participants compared their methods and approaches on standardized data depending on the chosen sub-challenge. A long-term goal was to gather the community to find common approaches and define conceptual strategies for a more accurate detection and analysis of RNA modifications. In addition, we intend to provide new impetus for developing new methods and improving data analysis and the comparability of methods.

## Results

The RMaP challenge focused on exploring and collecting methods for RNA modification detection and comparing methods of participating scientists on the same dataset (Fig. 1). The challenge was divided into three sub-tasks (Fig. 1c), where each of them requires detecting a different type of

modification. Each challenge was evaluated separately using the following metrics: root mean squared error (RMSE), mean absolute error (MAE), median absolute error (median AE), max and min deviations (see the "Methods" section for more details). Additionally, accuracy and F1-score were used to evaluate and compare the methods (Table 2).

### Challenge 1—Modification calling of 5-methylcytosine (m5C)

The first challenge consisted of detecting m5C modifications on RNA transcribed reads, where the modifications can occur at several unknown positions in the RNA sequence. An artificial DNA sequence was created to generate the data for this challenge. We generated two sets of reads from this template using in vitro transcription (IVT). Using either m5CTP or plain CTP, respectively, in the IVT reaction, the first set contained transcripts fully modified with m5C, and the second set correspondingly contained unmodified transcripts. Both sets were sequenced on an ONT MinION R9.4.1 flowcell. The resulting raw signals from both sets were then mixed into one dataset. This dataset was given to the participants in fast5 format together with the artificial DNA reference sequence in fasta format.

The participants did not know which read originated from which set of transcripts. The task of challenge 1 was to analyze the RNA reads and report the frequency of the specific m5C modification at the resolution of single nucleic bases per position in the DNA reference sequence. Target frequencies exist for 243 out of 2438 positions. The target frequencies range from 0.12 to 0.33. The results should be reported using the bedRmod file format (see Method Section for more details). The results of two methods (Method 1 and Method 2; see the "Methods" section for more information) were submitted for the challenge, shown in Fig. 2 and Table 2. Figure 2 shows that Method 1 has a smaller RMSE, Max. and Min. deviations values than Method 2, while accuracy and F1-sore are higher for Method 2. The MAE is comparable between Methods 1 and 2. This behavior in the results can be explained by the accuracy and F1-score calculation (see also metrics in the "Methods" section). In detail, these two metrics consider a range in positions (±one base) and frequencies to calculate as a positive result of one modification detection. This means that Method 2 predicts the expected value of frequencies in an acceptable range for 2.2% more positions than Method 1, within one base of the actual modification position (see metrics in the "Methods" section).

On the contrary, the 16% smaller RMSE value obtained by Method 1 indicates that Method 2 deviates more from the target frequency. Comparing the two pipelines on m5C analysis (see the "Methods" section), Bayespore combined with Dorado basecaller (Method 1) is more precise in detecting the correct frequency and has smaller values for max and min deviation. In contrast, CHEUI + Guppy basecaller (Method 2) predicts, on average, more modified positions correctly. The higher performance of Method 1 on frequency detection can also be linked and supported by the accuracy of basecalling from Dorado, which, according to ONT, is higher than Guppy. However, at this stage, it is impossible to assign the difference in performance between them only to the basecalling because also different resquiggle methods were used. Additional comparison is necessary to narrow down this hypothesis.

### Challenge 2—Modification calling of N6-methyladenosine (m6A)

Similar to challenge 1, the goal of challenge 2 was to detect RNA modifications in transcribed reads, in this case, m6A modification, which were incorporated into the RNA sequence. The data was generated the same way as in challenge 1, using a different artificial DNA genome designed for this modification. Again, participants were provided with raw RNA signals in fast5 format, both modified and unmodified, obtained from ONT sequencing, along with the artificial DNA sequence in fasta format. The task was the same as in challenge 1: to report the frequency of the specific m6A modification at the resolution of single nucleic bases on the DNA reference sequence and report the results using the bedRmod file format. Target frequencies exist for 243 out of 2438 positions. The target frequencies range from 0.01 to 0.1. In this case, results were submitted only for Method 3 (see the "Methods" section for more information). The results are shown in Fig. 3

## Table 1 | RNA modification detection tools for direct RNA sequencing data sequenced with ONT

| RNA detection method | Tested RNA modifications | Method approach | Ref. |
|---|---|---|---|
| nanoRMS/ nanoRMS2 | ψ, Nm, m6A | Direct & signal comparison | 41 |
| EpiNano | m6A, ψ, m2G, m7G, m3U | Signal comparison & error-profile | 43 |
| m6anet | m6A | Direct | 52 |
| Magnipore | any | Signal comparison | 44 |
| xPore | m6A | Signal comparison | 45 |
| Yanocomp | m6A | Signal comparison | 47 |
| Nanocompore | m6A, Ino, ψ, m5C, m62A, m1G | Signal comparison | 46 |
| ELIGOS | m6A, m1A, m5C, hm5C, f5C, m7G, Ino, ψ, 5moU | Signal comparison & error-profile | 39 |
| JACUSA2 | m6A | Signal comparison | 50 |
| Tombo | any | Signal comparison & direct | ONT |
| Nanom6A | m6A | Direct | 53 |
| DENA | m6A | Direct | 54 |
| mAFiA | m6A | Direct | 55 |
| Penguin | ψ | Direct | 56 |
| MINES | m6A | Direct | 58 |
| Nano-ID | e5U, Br5U, I5U, S4U, S6G | Direct | 59 |
| DRUMMER | m6A | Error-profile | 48 |
| DiffErr | – | Error-profile | 49 |
| CHEUI | m6A, m5C | Direct | 57 |
| NanoPsu | ψ | Direct | 60 |
| NanoSPA | ψ, m6A | Direct | 61 |
| TandemMod | m1A, m6A, m5C, m7G, hm5C | Direct | 62 |
| IL-AD | m1A, m6A, m5C, 5mC, 5hmC, ψ | Direct | 63 |
| nanoDoc | ψ, m7G, m5C, Cm, Gm, m6A, m1A, m2G, m5U … | Direct | 51 |
| ModiDeC | m6A, ψ, Ino, Gm, m1A | Direct | 64 |

Direct approaches take only one sample as input to predict modifications. Comparative approaches, as well as error-profile analysis, take two samples as input, typically a modified sample compared to an unmodified control.

and Table 2. Even though only Method 3 was successfully submitted, this can be compared qualitatively to the methods of challenge 1, as the data were generated similarly. In detail, we can compare Method 2 and Method 3, both of which use the same pipeline (see the "Methods" section), now used for m6A prediction. We can observe that RMSE, MAE, min., and max. deviation values are lower for Method 2 compared to Method 3, but accuracy and F1-score are higher for Method 2. While the accuracy is slightly worse for Method 3, the $F_1$-score is way lower, which can be caused by a large number of false positive predictions, false negative predictions, or a combination of both. These results suggest a lower error in frequency values and position detection on average when this method is used for m5C analysis compared to m6A. However, the expected modification frequencies are lower for the m6A dataset than for the m5C dataset. This may lead to a more difficult analysis due to a lower range of values for predicting the correct frequency. Overall, CHEUI seems to be better at predicting the m5C modification frequency than the m6A frequency in the given dataset due to the lover RMSE and MAE obtained in Method 2.

### Challenge 3—Machine learning training and modification calling of Pseudouridine (Ψ)

Challenge 3 focused on detecting pseudouridine Ψ in RNA reads using machine learning techniques. As in the other two challenges, an artificial DNA sequence was created, and IVT was used to generate two sets of reads, one containing fully modified Ψ reads and one with unmodified reads. Both sets of RNAs were sequenced with ONT. The reads were then mixed into one dataset. This dataset was split into two subsets (80% for training and 20% for testing). Thus, both sets contained unmodified RNA sequences and reads with Ψ modifications. Both datasets in fast5 format and the used DNA reference sequence in fasta format were given to the participants.

In addition, it was indicated which positions in which read in the training data set were changed. Herewith, the participants could generate and train their own machine learning method for RNA modification detection, which requires labeled data for training. The task of challenge 3 was to analyze the test dataset, in which only the raw signal was given in form of fast5 files, but nothing was known about the modification positions. Target frequencies existed for 243 out of 2438 positions. The target frequencies range from 0.31 to 0.5. Four methods (Methods 4–7) competed in this task, and the results are shown in Fig. 4 and Table 2. Methods 4 and 5 show similar results, especially for accuracy, F1-score, and min deviations. On one hand, Method 4 presents lower values for RMSE and MAE, which, combined with the high values of F1-score and accuracy, indicates that this method more accurately predicts both the frequency and the position. However, on the other hand, Method 4 relies on pre-conversion of all Us to Cs in the reference sequence (as pseudouridine is known to cause a strong U-to-C basecalling error), implying that prior knowledge on the modification type in question is required to implement Method 4, compared to Method 5, which does not require prior knowledge of the modification type that is meant to be identified (i.e. pseudouridine in this case). The very small values for RMSE and MAE point out that both approaches come close to the ground truth, with slightly lower values obtained by Method 4. Important to point out is that Method 5 shows a smaller value for the max deviation (about 23% less). Comparing the pipeline used in this challenge (see Methods 4–7 in the "Methods" section), the difference in performance can suggest that the post-basecalling and alignment pipeline significantly impacts the analysis. All four methods use Dorado or Guppy combined with minimap2, but Methods 4 and 5 use different basecaller, and both have similar performance. This suggests that using Dorado or Guppy basecallers has a minor impact on the analysis, although they perform differently, if the basecaller has a sufficiently accuracy threshold for nucleotide detection. After these steps, the pipelines begin to diverge. However, an interesting aspect is that Methods 5 and 6 both use gradient boost, but they have a different pipeline for signal-to-base association. This difference between the two methods seems to reduce the values of RMSE and MAE, suggesting an essential passage in developing pipelines. This is also supported by Method 4, which has low RMSE and MAE values using Tombo and deep learning for the analysis.

## Discussion

The RMaP challenge aimed to address the problem of identifying RNA modifications from DRS raw signal by giving a specific task for a particular problem to analyze one type of RNA modification at a time. From the comparison of the several methods of each task, we can establish that Methods 1, 3, and 4 were the winners of Challenges 1, 2, and 3, respectively (Table 2). However, the other methods were also competitive and occasionally outperformed the winner in one parameter or the other, showing, not unexpectedly, that highly performant solutions are yet to be developed, leading to the question of what we can learn from the several methods reported in this work. To address this question, it is possible to compare the methods pipeline to interpret some aspects, also if they participate in different tasks, and see if common trends can be underlined. Methods 1 and 4 were the winners of Challenges 1 and 3, respectively (Table 2), and both methods used Dorado basecaller from ONT in their pipeline. However, they

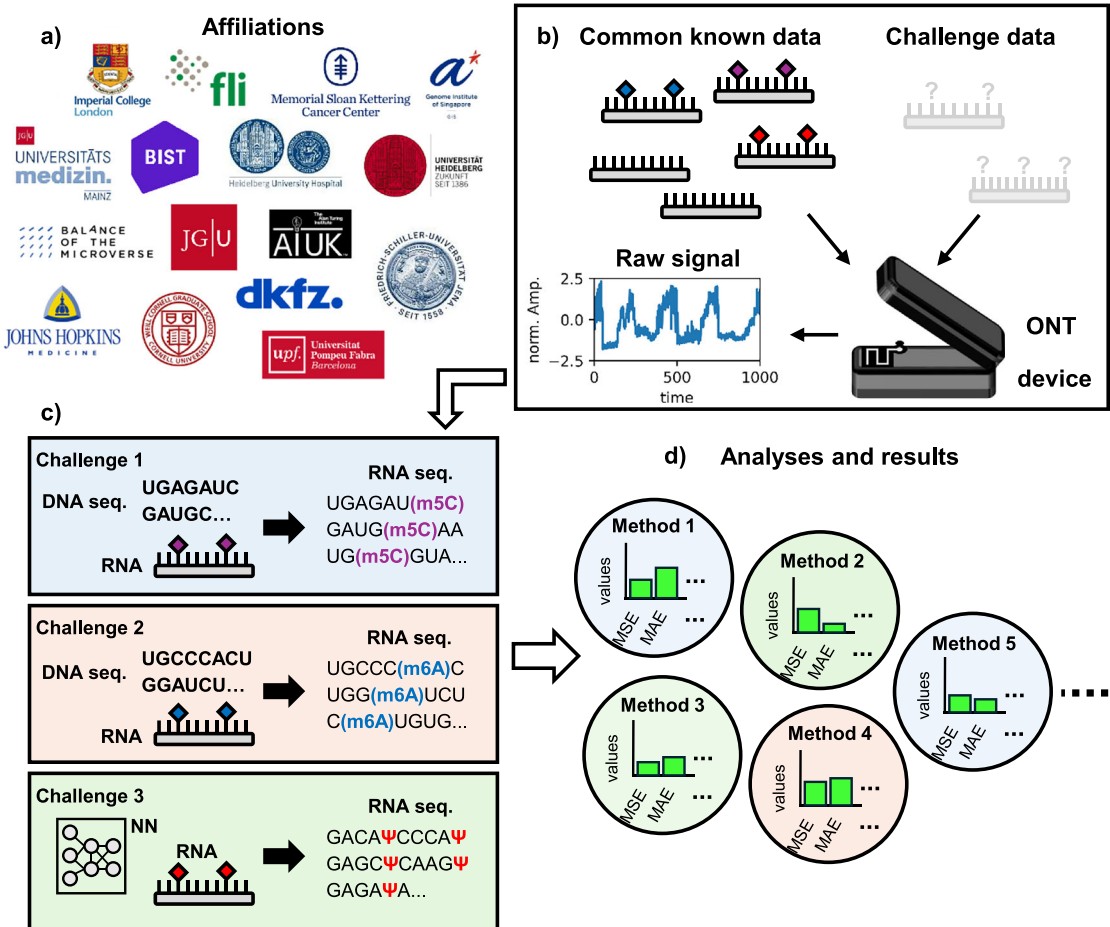

**Fig. 1 | The RMaP challenge workflow.** RMaP challenge overview. **a** The several affiliations that contributed to the RMaP challenge. **b** Data preparation pipeline for each sub-challenge in RMaP. Datasets were prepared in vitro and measured with ONT. **c** General overview of the three sub-challenges proposed in RMaP. Each of them proposed a different task for selected RNA modifications. **d** The results obtained by the new methods are analyzed and compared.

**Table 2 | Values of each metric for each method submitted to the RMaP challenge obtained by comparing the method's predictions with expected values**

| Metrics | Challenge 1 | | Challenge 2 | Challenge 3 | | | |
|---|---|---|---|---|---|---|---|
| | **Methods** | | **Methods** | **Methods** | | | |
| | 1 | 2 | 3 | 4 | 5 | 6 | 7 |
| RMSE | **0.052** | 0.062 | **0.105** | **0.010** | 0.021 | 0.145 | 0.148 |
| MAE | **0.015** | **0.015** | **0.033** | **0.002** | 0.006 | 0.046 | 0.047 |
| Max dev. | **0.550** | 0.666 | **0.809** | 0.142 | **0.109** | 0.497 | 0.653 |
| Min dev. | <0.001 | <0.001 | <0.001 | <0.001 | <0.001 | 0.309 | 0.309 |
| Accuracy | 0.938 | **0.960** | **0.820** | **1.000** | **1.000** | 0.822 | 0.900 |
| $F_1$ | 0.554 | **0.750** | **0.164** | **1.000** | **1.000** | 0.822 | 0.900 |

Metric values are calculated using the formula given in the "Methods" subsection "Metrics". The best values for each metric and challenge are marked in bold.

used different resquiggle algorithms (Remora and Tombo) to assign the RNA raw signal to the corresponding base. This suggests that the resquiggle algorithm has a minor impact on the pipeline during data generation and labeling compared to the basecaller. This is also supported by the fact that Methods 6 and 7, which both use Guppy during data analysis, have an overall worse performance in each metric field compared to Method 4 (Table 2). We can tentatively interpret that it is an advantage to use the same common initial point, which is to combine Dorado with any of the resquiggle-methods reported here.

Another aspect that can be understood by comparing the methods proposed here in the RMaP Challenge is the importance of selecting the correct prediction algorithm. This can be deduced by comparing Methods 2 and 3, which both have CHEUI as the core algorithm for prediction but different RNA modification analyses. All metrics values are lower when CHEUI is used for the m5C dataset analysis compared to m6A dataset, which suggests that CHEUI is more suitable and specific for m5C analysis. A similar conclusion can also be obtained if we compare Methods 4 and 7 in Challenge 3. Both methods use a deep learning method to detect Ψ

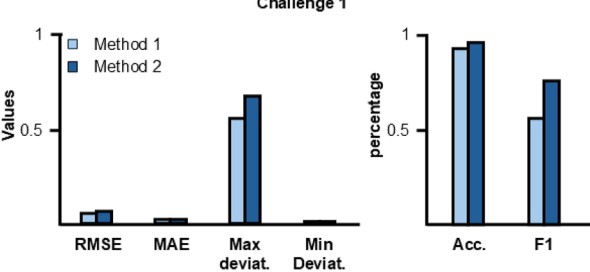

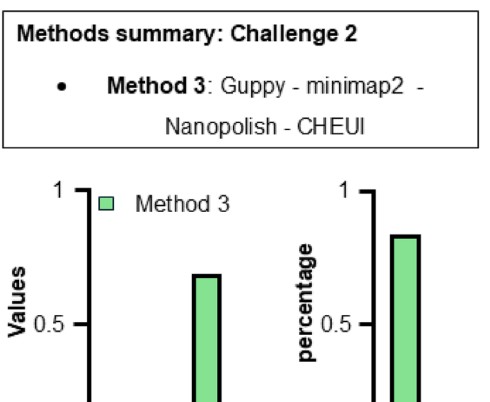

**Fig. 2 | Method summary and results of RMaP challenge 1.** (Top) Summary of methods used for challenge 1. (Bottom) Comparison between Methods 1 and 2 performances on $m^5C$ modification detection. A lower value is better for all metrics. The diagram shows the values of each metric used in this work. The metrics values were obtained by comparing the two methods predictions with expected values. Metric values can be found in Table 2.

**Fig. 3 | Method summary and results of RMaP challenge 2.** (Top) Summary of methods used for challenge 2. (Bottom) Method 3 performance on $m^6A$ modification detection. A lower value is better for all metrics. The diagram shows the values of each metric used in this work. Metric values can be found in Table 2.

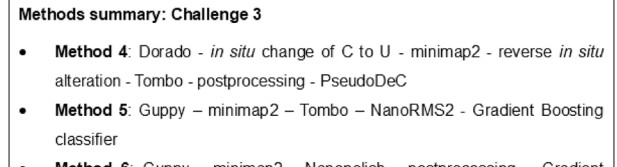

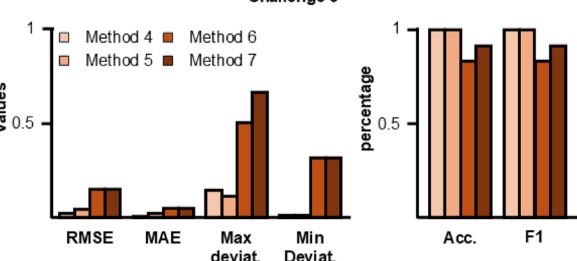

**Fig. 4 | Method summary and results of RMaP challenge 3.** (Top) Summary of methods used for challenge 3. (Bottom) Comparison between methods 4–7 performances on Ψ modification detection. The graph shows the values of each metric used in this work. The metrics values were obtained by comparing methods predictions with expected values. Metric values can be found in Table 2.

underlining the delicate aspect of data creation and selection to design new challenges. This is highlighted, for example, by the $F_1 = 1$ for Methods 4 and 5, which suggests a high performance on the specific problem analyzed here during the RMaP challenge, but they may miss general patterns in real biological data. This is because generating ground truth data sets that can mimic high complexity biological samples can be challenging itself. In fact, algorithms can have discrepancies in performance when used for synthetic or biological RNA data set analysis[65], pointing out that for further challenges the combination of synthetic (including IVT) and in vivo DRS RNA samples can be used for training and validation respectively. This approach can highlight new algorithms strategies for the analysis of biological samples. However, this suggests that benchmarked biological DRS data should be used, underlining the importance of generating RNA databases where RNA modifications are established[66–68]. This also suggests that many more challenges are still required in order to more precisely define a guideline for the analysis of RNA modifications on ONT data. Still, these can only help the complex field of epitranscriptomics in the future. We also propose the development of a comprehensive library of (a) synthetically produced and specifically modified RNAs of different sizes and modification patterns and (b) in vivo datasets of different RNA species. This will also enable simple and comparable benchmarking of instruments for detecting RNA modifications.

## Conclusion

The RMaP challenge aimed and still aims at bringing together the community to jointly address the identification of RNA modifications from DRS raw signals in a fashion that was both comparative and competitive. As such, it was an experiment in itself, and future assessments may show its effect on community shaping. Meanwhile, ONT has transitioned to the new pore (RP4) for RNA sequencing together with the sequencing kit (SQK-RNA 004, 'RNA' Flowcell)[56] and an obvious question of burning interest is, if the conclusions reached in this work can be applied to the new "chemistry." We are, therefore, currently exploring options for another challenge in this backdrop.

## Methods
### IVT design

The reference DNA sequences used for the challenges were specifically designed to include all possible 5-mers, each containing a single

modification, but with the difference that PseudoDec and m6Anet were developed to detect Ψ and $m^6A$, respectively, resulting in a significant difference in performance during the analysis. The difference in performance cannot be associated with deep learning per se but rather with the specificity of the neural network architecture designed for a specific task. Another critical aspect that can be captured by comparing the methods is the importance of feature extraction, which correctly links the raw signal to a possible modified basis. This can be deduced by closely comparing Methods 5 and 6, which both use Guppy and Gradient Boosting classifiers but use different approaches for feature extraction, obtaining very different values in performance. This, for example, can be observed in the MRSE and MAE values, which are much lower for Method 5 than Method 6.

By combining the results of this challenge, we can suggest a standard guideline for RNA modification detection. This means e.g., using any type of resquiggle algorithm, as long as it is used with Dorado basecaller. Furthermore, it is of critical importance to carefully select the correct prediction algorithm for the analysis. However, these are only suggestions from preliminary results obtained from synthetic data and not in vivo RNA analyses,

modification at the central position. Each reference sequence comprises 256 unique 5-mers arranged sequentially without any intervening spaces or gaps between them. This design ensures a continuous sequence that facilitates the analysis of modified bases while maintaining a compact and systematic representation of all 5-mers.

## Data generation

For production of unmodified and modified RNAs, the synthetic DNA templates were ordered as double-stranded DNA Fragments (GeneStrands —eurofins Genomics) containing the sequences of all possible 5-mers with a central C, A, or T. For each template a modified and unmodified dataset was produced. 200 ng of double stranded DNA template were used in 20 μl IVT reactions for 1 h using the HighYield T7 RNA synthesis Kit (Jena Biosciences—RNT-201), following the manufacturer's instructions. In some reactions single standard nucleosides (CTP, UTP, ATP, GTP included in the kit) were replaced by respective modified nucleosides (Jena Bioscience— NU-1138S 5-Methyl-CTP, NU1139S Pseudo-UTP, NU-1101S N6-Methyl-ATP). After IVT the DNA templates were digested by addition of 80ul RNase-free water including 2U of DNase I (NEC—M0303S) and 10x DNAse reaction buffer for 15 min at 37 °C. The RNA products were purified using the RNAClean XP beads 1:1 volume IVT/DNase reaction mix to bead volume (Beckman Coulter—A66514) following the manufacturer's instructions. The cleaned RNA was measured by Qubit RNA-HS (High Sensitivity)-Assay-Kit (ThermoFisher Scientific—Q32852) accordingly to the manufacturer's instructions. Finally, a poly A-tail was added to the RNAs by incubation of 1 μg RNA with 1 mM ATP and 5U *E. coli* Poly(A) Polymerase (NEB—M0276) for 30 min at 37 °C. The Poly-A tailed RNA products were again purified using the RNAClean XP beads 1:1 volume RNA reaction mix to bead volume (Beckman Coulter—A66514) following the manufacturer's instructions. The cleaned RNA was measured by Qubit RNA-HS (High Sensitivity)-Assay-Kit.

RNA sequencing was performed following the instruction provided by Oxford Nanopore Technologies (Oxford, UK), using R9.4 chemistry flowcells (FLO-MIN106) and direct-RNA chemistry sequencing kit (SQK-RNA002). For library preparation we used 500 ng of pooled poly-A tailed IVT RNA templates, prepared as described above, using the provided polyT (RTA) adapter.

## Determining modification target rates

For each challenge, we sequenced two distinct sets of reads: one containing unmodified sequences and the other fully modified with a single type of modification. Reads were base called using the dorado_basecall_server software package (v7.1.4) from ONT with the r9.4.1 hac model. To determine positional target modification rates, we first mapped the reads separately, before mixing the datasets. To map the reads, we used the ont_aligner from the same software package. The ont_aligner utilizes minimap2 (v2.24) with custom parameters preset by ONT. Our commands can be found in the supplements. After mapping, for each position in the reference sequence, we counted the number of reads covering that position in both datasets and calculated the modification ratio by comparing the counts between the two sets on a per-position basis. Finally, we combined the datasets for each challenge to ensure participants could not distinguish the origin of individual reads, preserving the blind nature of the analysis.

## Data preparation commands

RMaP challenge data sets were prepared using the following commands line in the prompt:

1. ont_basecaller_supervisor --input_path "$input" --save_path "$save_path" --config rna_r9.4.1_70bps_hac.cfg --disable_qscore_filtering --disable_pings
2. ont_aligner -i "$basecalls" -s "$out" --align_ref "$reference" --bam_-out --alignment_filtering none --minimap_opt_string -k5 --minimap_opt_string -x"map-ont"where the "$ + name" is a variable to fill to run the prompt. For example, "$input" indicates to introduce the input path name.

## Metrics

To evaluate the methods proposed in this challenge, we use several metrics that include root mean squared error (RMSE), mean absolute error (MAE), median absolute error (median AE), max and min deviations. We have a set of $N$ observed values for each task and submission, $Y_i$, and a matching set of predicted values $\hat{Y}_i$. The formula of each metric is reported here below:

$$RMSE = \sqrt{\frac{1}{N} * \sum_{i=0}^{N} \left(Y_i - \hat{Y}_i\right)^2} \qquad (1)$$

$$MAE = \frac{1}{N} \sum_{i=0}^{N} (|Y_i - \hat{Y}_i|) \qquad (2)$$

Min and max deviation between observed values $Y_{\mathrm{mod}\_i}$ and predicted values $\hat{Y}_{\mathrm{mod}\_i}$ on the modified positions were also calculated to give an error range on the predicted modification frequencies.

$$\mathrm{max}\ deviation = \max_i(|Y_i - \hat{Y}_i|) \qquad (3)$$

$$\mathrm{min}\ deviation = \min_i(|Y_i - \hat{Y}_i|) \qquad (4)$$

$F_1$ score, which combines the precision and recall in one metric, and accuracy values were also considered in the metrics evaluation process, and they are calculated as follows:

$$Acc. = \frac{TP + TN}{N} \qquad (5)$$

$$F1 = \frac{2 * TP}{2 * TP + FP + FN} \qquad (6)$$

where $N$ is the total number of expected values, TP and TN are the true positive and negative, respectively, while FP and FN are the false positive and false negative, respectively. For each modified position with a given target frequency $Y$, the frequency is correctly predicted, if the prediction $\hat{Y}$ is within a range of $Y \pm Y * 0.6$ and $\pm 1$ base position from the expected one (TP, else FN). For unmodified positions with a target modification rate of 0, the prediction is correct if it is also 0 (TN, else FP).

## Pipeline description: Method 1

The fast5 files were first converted to POD5 format using pod5tool (v0.2.4, pod5 convert fast5), followed by basecalling with Dorado (v0.3.4). The basecalled reads were then aligned using minimap2 (v2.26). The reference kmer table for rna_r9.4_180mv_70bps was downloaded from the ONT GitHub repository (https://github.com/nanoporetech/kmer_models). Next, bayespore was run with the POD5 and BAM files, the kmer table, and other default parameters as inputs.

## Pipeline description: Methods 2 and 3

The raw data in fast5 format was base called using Guppy (v6.5.7+ca6d6af) with default parameters and rna_r9.4.1_70bps_hac base calling model. The reads passing quality filter in fastq format were aligned to the reference genome using minimap2 (v2.24-r1122) with following parameters (-ax map-ont -uf -t 48 -N 20). Signal values for each 5-mer in the reads was generated using eventalign module of nanopolish (v0.14.0) and kmer level signal information was generated. The signal information was used with kmer models generated by CHEUI to pre-process the data for identification and calculation of m6A and m5C modifications frequencies. The preprocessed files were used to predict site level m6A and m5C modifications on the reference genome provided.

## Pipeline description: Method 4

The RNA fast5 files were basecalled using dorado basecaller (v0.3.2) provided by ONT. The RNA 0002 high accuracy model was used to basecall the

```
#fileformat=bedRModV1.8
#organism=9606
#modification_type=RNA
#assembly=GRCh38
#annotation_source=Ensemble
#annotation_version=93
#sequencing_platform=Illumina NovaSeq 6000
#basecalling=post_basecalling.cfg
#bioinformatics_workflow=https://github.com/anambu/bedRMod
#experiment=https://doi.org/test
#external_source=GEO;GSETEST
#methods=TEST
#references=pubmed_id:12345678

#chrom  chromStart  chromEnd  extraColumn  name  score  strand  thickStart  thickEnd  itemRgb       coverage  frequency
1       999         1000      50           m1A   900    +       999         1000      100,128,128   60        90
2       1000        1001      60           m5C   930    -       1000        1001      0,139,139     90        83
X       3000        3001      70           m3C   920    -       3000        3001      128,128,128   12        39
Y       4000        4001      80           m3C   920    -       4000        4001      128,255,0     10        37
MT      5000        5001      90           m3C   920    -       5000        5001      128,255,0     46        65
```

**Fig. 5 | Example bedRMod file.** Text visualization of a bedRMod file.

RNA reads and the dorado analysis was stored as fastq. To achieve an alignment of about 80% using minimap2[69] (k-mer size = 8, -ax ont-map flag), every uridine/thymine nucleotide was substituted with a cytosine in the fastq files and the fasta reference[70]. To underline, the alignment without the substitution was less than 5%. After alignment, the substituted cytosine in the fastq files and fasta were reversed back to uridine/thymine. The aligned dataset was then resquiggling using Tombo from ONT, which was used to associate specific raw signal to their respective basecalled nucleotide. Next, the processed data were used to train PseudoDec (https://github.com/mem3nto0/PseudoDeC_RMaP) which is a neural network that can be trained using processed data from Tombo for modification detection. The deep neural network will then analyze the raw signal and its respective sequence to remap all the sequence, pointing the modification position and type. For the challenge, the neural network was trained for pseudouridine detection.

### Pipeline description: Method 5
This pipeline uses the nanoRMS2 methodology[71] with minor modifications, described below. Firstly, the reads were basecalled with Guppy (v6.0.6) using the RNA002 hac model and storing trace information with --fast5-out. Subsequently, reads were aligned with minimap2 (v2.26) and resquiggled with Tombo (v1.5). Then, a set of features (signal intensity, dwell time, trace, modification probability) was stored in a BAM file for every base for every read using the get_features.py script, which is part of nanoRMS2. One Gradient Boosting classifier (as implemented in scikit-learn) was trained for every 3-mer centered at T using reads from the training set. Finally, trained classifiers were used to predict modification status of T positions in the reads from the testing set. Additional details and code are available at https://github.com/novoalab/RMaP_challenge.

### Pipeline description: Method 6
The sequencing reads were basecalled using Guppy (v6.4.2) provided by ONT. Following basecalling, individual fastq files were merged into a single fastq file, and an index was generated using the index module from Nanopolish[72] (v0.14.0). Alignment of the reads to the synthetic reference sequence was performed using Minimap2[57] (v2.24), with the parameters set to -ax splice -uf -k14. The resulting mapped reads were sorted, indexed, and converted into BAM files using SAMtools[73] (v1.5). To align the nanopore signal squiggles to the reference genome and extract per-site features, the Nanopolish eventalign module was utilized. Due to the large disparity between the unmodified and modified data, 0.05% of the unmodified data was randomly sampled, resulting in approximately 50,000 data points, which were integrated with the modified dataset. In accordance with the challenge guidelines, every 5th position in the modified data was labeled as containing a modification, assigning these positions to class 1, while the unmodified positions were designated as class 0. The signal was vectorized by applying the signature transform from rough path theory, enabling the extraction of key features from the sequential data. These transformed features were used to construct feature vectors, which captured the essential characteristics of the signal. Finally, a Gradient Boosting algorithm from scikit-learn[74] was applied to these feature vectors to predict the locations of modifications in the DRS signal. The model utilized the enriched feature representation from the signature transform to improve predictive accuracy.

### Pipeline description: Method 7
The training fast5 files were split into two groups: unmodified and modified reads. Fast5 files from both groups, along with those from the test set, were basecalled separately using Guppy (v6.5.7) to generate fastq files. These fastq files were then aligned with minimap2, using the parameters -ax splice --secondary=no -k5, and the resulting sam files were converted to bam format using samtools. Next, the fast5, fastq, and bam files were processed with f5c eventalign, applying the parameters --signal-index and --scale-event for event segmentation. The eventalign.txt files generated by f5c were processed with m6anet dataprep, modified to output NNTNN (equivalent to NNUNN) kmers. After merging the datapreps from the unmodified and modified groups, each site was labeled as either modified or unmodified. The merged dataprep files (data.json and data.info) were used to train m6anet. Finally, the trained model was used for predictions on the test data through m6anet inference.

### Data format
The bedRMod format is a unified data format for storing RNA modification data to enable sharing, collaborating and reuse of data, greatly enhancing the speed at which research can be done. It is based on the standard browser extensible (BED) format[73], a text file format with tab-delimited rows. Introducing the bedRMod format for storing epitranscriptomic data in the RMaP Challenge creates a basis for comparable results across different methods of detecting RNA modifications. This is especially the case as a uniform data format for storing epitranscriptomic data does not exist, yet. bedRMod provides a new format, which is compatible with many established tools and thus easy to adapt into already existing workflows.

A bedRMod file consists of two main parts: The header which contains metadata, clarifying from where the RNA modification data originates and how it was obtained and the data section which stores the site-specific modification data. In the data section each row contains the site-specific RNA modification properties of one modification at one position. An example of the structure a bedRMod file can be seen in Fig. 5. For the complete specification of bedRMod, please refer to: github.com/anmabu/bedRMod/blob/main/bedRModv1.8.pdf. The advantage of using bedRMod over other formats is that it was specifically designed to be used with epitranscriptomic data. Additionally, it is straightforward to use bedRMod, as it can be viewed with any text editor and due to its extensive header, the contents are easy to interpret. A toolkit for conversion of existing RNA modification data into bedRMod was implemented using python3.10. It can be found at https://github.com/anmabu/bedRMod. A graphical user interface (GUI) is also available for ease of use.

## Data handling and storage

All aspects of data management were provided through a NextCloud installation, the RMaP Challenge Cloud, which was exclusively implemented for this benchmark event by the Dieterich Lab in Heidelberg. Two virtual machines with 4 GB RAM each and 200GB shared disk space were dedicated to this purpose. For organizational reasons, we then set up a predefined folder structure for handling outgoing data ("challenge data"). Incoming data, i.e. "challenge solutions" were uploaded to private folders by challenge participants. Use and access privileges were managed through LDAP and implemented to meet the needs of data owners, solution providers and data managers. Instructions, guidelines and specifications were deposited in the RMaP Challenge Cloud as well. We performed community briefings with references to all information material by sending round mails.

## Code availability

The code used for the RMaP challenge can be found on the several GitHub pages here listed:• Method 1: https://github.com/chilampoon/bayespore. • Methods 2 and 3: https://github.com/comprna/CHEUI. • Method 4: https://github.com/mem3nto0/PseudoDeC_RMaP. • Method 5: https://github.com/novoalab/RMaP_challenge. • Method 6: https://github.com/jts/nanopolish and https://scikit-learn.org. • Method 7: https://github.com/GoekeLab/m6anet

## Data availability

The RMaP challenge DRS fast5 files are stored in the European Nucleotide Archive (ENA) and they can be found using the following Project Accession ID: PRJEB84053.

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

## Acknowledgements

This work was supported by the DFG (German Research Foundation: TRR-319 TP C01, Project Id 439669440 to M.H., and CRC 1076 "Aqua-Diva" TP A06). S.G. and N.A. acknowledge funding from the Forschungsinitiative Rheinland-Pfalz and the ReALity initiative of the Johannes Gutenberg University Mainz. V.D. and S.G. acknowledge funding by SFB 1552 Project No. 465145163 of the Deutsche Forschungsgemeinschaft (DFG).

## Author contributions

Jannes Spangenberg designed and measured the modified synthetic RNA using DRS. He also designed the pipeline for generating the RMaP challenge database and wrote the manuscript. Submission solution and data analyses were performed by Jannes Spangenberg under the supervision of Nicolò Alagna. Nicolò Alagna, Mark Helm, Susanne Gerber, Manja Marz and Christoph Dieterich designed and supervised the study and revised the manuscript. All authors have given approval to the final version of the manuscript.

## Funding

## Competing interests

The authors declare no competing interests. Mark Helm is a consultant for Moderna Inc.

## Additional information

[1]RNA Bioinformatics, Friedrich-Schiller-University Jena, Leutragraben 1, 07743 Jena, Germany. [2]Institute of Pharmaceutical and Biomedical Sciences, Johannes Gutenberg-University Mainz, 55128 Mainz, Germany. [3]Institute for Informatics, Johannes Gutenberg-University Mainz, 55128 Mainz, Germany. [4]Institute for Human Genetics, University Medical Center of the Johannes Gutenberg University Mainz, Mainz, Germany. [5]Fritz Lipmann Institute-Leibniz Institute on Aging, 07745 Jena, Germany. [6]Centre for Genomic Regulation (CRG), The Barcelona Institute of Science and Technology, Dr. Aiguader 88, Barcelona 08003, Spain. [7]Universitat Pompeu Fabra, Barcelona 08003, Spain. [8]ICREA, Pg Lluis Companys 23, Barcelona 08010, Spain. [9]Department of Neurosurgery, Oncology, Sidney Kimmel Comprehensive Cancer Center, School of Medicine, Johns Hopkins University, 1650 Orleans St, Baltimore, MD 21231, USA. [10]Johns Hopkins All Children's Hospital, 600 5th St. South, St.Petersburg, FL 33701, USA. [11]Division of Immune Diversity, German Cancer Research Center (DKFZ), 69120 Heidelberg, Germany. [12]Department of Mathematics at Imperial College London, London SW7 2AZ, UK. [13]The Alan Turing Institute, London NW1 2DB, UK. [14]Graduate Program of the Faculty of Biosciences, Heidelberg University, Heidelberg 69120, Germany. [15]Computational Oncology, Memorial Sloan Kettering Cancer Center, New York, NY, USA. [16]Department of Physiology and Biophysics, Weill Cornell Medicine, New York, NY, USA. [17]Genome Institute of Singapore (GIS), Agency for Science, Technology and Research (A*STAR), Singapore 138672, Republic of Singapore. [18]Department of Statistics and Applied Probability, National University of Singapore, Singapore, Republic of Singapore. [19]Klaus Tschira Institute for Integrative Computational Cardiology, University Hospital Heidelberg, Im Neuenheimer Feld 669, 69120 Heidelberg, Germany. [20]Balance of the Microverse, Fürstengraben 1, 07743 Jena, Germany. [21]Institute for Quantitative and Computational Biosciences (IQCB), Mainz, Germany. ✉e-mail: christoph.dieterich@uni-heidelberg.de; mhelm@uni-mainz.de; manja@uni-jena.de; sugerber@uni-mainz.de; nalagna@uni-mainz.de

