## [Peer Review file · Communications Chemistry]

Predicting RNA modifications by nanopore sequencing: The RMaP challenge

Corresponding Author: Dr Nicolò Alagna

Version 0:

Reviewer comments:

Reviewer #1

(Remarks to the Author)

This paper reports on the RMaP challenge, which aims to compare methods for RNA modification detection. However, the evaluation is limited to the IVT dataset generated in this challenge. Independent IVT datasets should be systematically evaluated to assess the robustness of these methods. Additionally, the manuscript lacks essential information that should be provided before it can be considered for publication.

Major issue:

Eighty percent of the dataset was used for method training, with predictions based on the remaining 20%. The predictive accuracy of these methods should also be evaluated on independent IVT datasets, such as Curlcake, ELIGOS, or IVET. Additionally, the top-performing methods should be tested in real-case scenarios using in vivo DRS datasets.

Other issues:

All the DRS data is missing, which should be publicly available.

The reference sequences as templates for in vitro transcription should be provided.

The methods of in vitro transcription, DRS library construction and ONT sequencing should be provided.

The github https://github.com/novoalab/RMaP_challenge is not available.

In figure 1C, Challenge 2, Whether is Genome seq or RNA seq?

Reviewer #2

(Remarks to the Author)

Spangenberg et. al. describe a community challenge to detect three different RNA modifications from Oxford Nanopore Technologies' direct RNA nanopore sequencing (dRNA-seq). They generated three synthetic datasets using in vitro transcription (IVT) where they replaced a canonical rNTP with m5C (challenge1), m6A (challenge2), or pseudouridine (challenge3) during synthesis. They also synthesized the fully canonical IVT RNA transcripts for each challenge and sequenced all 6 samples independently on RNA002 MinION flowcells. The data were mixed in silico and presented to the community with minimal details regarding the sample composition for them to attempt to detect the RNA modifications like they would approach a real biological sample. There were seven submissions across all three challenges with the f1 scores ranging between 0.16 and 1.0. The authors compared the analysis pipelines and concluded that the basecalling version has a larger impact on accurate RNA modification detection than any other observed difference. Although the authors generated data using the now discontinued RNA002 chemistry, they propose to organize a second RMaP challenge using the newer RNA004 dRNA-seq chemistry from ONT.

Although the details on the challenge data are described in the results section, there is no mention on how they were generated in the methods section. Details on the IVT reactions, library preparation, sequencing conditions, and software versions would help in evaluating the conclusions drawn from these challenge data and replicating the results from others in the community. Supplementary figures on the sample QC for the three challenge datasets would also help to interpret the results generated by the seven submissions. Furthermore, details regarding how the ground truth of the modification ratios were established would help understand how the tools performed. Were the ratios of modified to unmodified established using the same basecaller and alignment versions as the submissions, or were they determined by one set of basecaller and alignment conditions? There may be alignment differences between reads called with guppy or dorado, and the different versions of minimap2 can alter the alignments, especially near the ends of reads. It is unclear if the positional modification ratios were established on a case-by-case basis, or predefined using one set of bioinformatics conditions.

Accuracy and F1 scores of 1.0 are very very good. I worry that the PseudoDeC and the Gradient Boosting classifier are overfit to the training data might perform worse on an IVT dataset with a different genomic sequence or a real biological sample. This is a trend we've seen before where the F1 scores are high in synthetic oligos or IVT datasets, but much lower in biological samples (<https://doi.org/10.1038/s41467-023-37596-5>; <https://doi.org/10.1093/bib/bbae001>). This is likely due to how difficult it is to establish an accurate ground truth set from biological samples and highlights how important the underlying data are in interpreting the results from tests like this. I think that the discussion could spend more time highlighting how challenging data generation is for these types of challenges and that there is a need for highly characterized samples, akin to genomes in a bottle, but for RNA modifications.

Given that most of the RNA modification detection tools are designed to detect m6A, I was surprised that only CHEUI made a submission for challenge2. Do the authors have any insights as to why there were so few submissions for this challenge and why the results in this challenge differ so substantially from the results seen in the paper describing CHEUI's development and deployment (<https://doi.org/10.1038/s41467-024-47953-7>)?

Minor Notes:

Table 1 doesn't really seem necessary for the manuscript. There have been a few different review articles that have summarized the different tools and only 2 of the 19 tools in the table submitted to the challenge. Furthermore, in addition to m6A, Nanocompare was tested to detect inosine, m5C, pseudouridine, m62A, m1G, and 2'OMeA using synthetic oligos. ELIGOS was tested not only to detect m6A, but also m1A, m5C, hm5C, f5C, m7G, inosine, pseudouridine, and 5moU using IVT constructs. Also, nanoDoc was tested on "23 different types of modifications in *Escherichia coli* and *Saccharomyces cerevisiae*" rRNA.

I think table 2 would be more useful to appear in the document before figure 2

Reviewer #3

(Remarks to the Author)

The manuscript titled "Predicting RNA modifications by nanopore sequencing: The RMaP challenge" aimed to improve algorithms' comparability, reliability, and consistency in RNA modification prediction. My major concerns are as follows:

The main purpose of RMaP challenge is not so clear to me. To compare different pipelines using current algorithms? To develop new state-of-art algorithms?

Authors set "to predict target frequencies" as goal of challenges. Which may over-state the power of method due to balance of false positive and false negative.

Table 1 is not sufficient since many algorithms are missing here, espacilly these one-to-all methods. Publication information or download links should be included.

Authors linked "the higher performance of Method 1 on frequency detection" with "the accuracy of basecalling from Dorado". Which is less interested since the focus of challenges should be the modification prediction. Which also makes the comparison between "Bayesore" and "CHEUI" not informative.

Authors also suggested "that the post-basecalling and alignment pipeline significantly impacts the analysis." (Line 261) Which also makes the comparison uninformative.

A controversial statement appeared in Line 264, "suggests that the basecaller has a minor impact on the analysis.", we do not know.

Meanwhile, different vesions of Dorado (v0.3.4 & v0.3.2), Guppy (v6.0.6, v6.4.2 & v6.5.7) & minimap2 (v2.26 & v2.26) were employed during analysis. With or without documented parameters. Check the citations of minimap2.

Due to the comparability issues among methods, the discussion and conclusion section are short of solid supports from the results.

There are still some non-negligible minor issues:

Fig1. Affiliation/Institute contributions are missing. Author contributions are missing.

Line 172, Line 210 & Line 246, authors wrote "The target frequencies range from 0.12 to 0.33.", "The target frequencies range from 0.01 to 0.1." & "The target frequencies range from 0.31 to 0.5." Did authors tune the target frequencies by adjust the mix ratio of modification data and plain data?

Line 219, authors wrote "which is caused by a large number of false positive predictions." I guess not only false positive, but also false negative.

Line 233, "an artificial DNA sequence", is it a random sequence or a thoughtful sequence for purpose?

Version 1:

Reviewer comments:

Reviewer #1

(Remarks to the Author)

The authors have addressed most of my previous comments.

Reviewer #2

(Remarks to the Author)

The authors addressed most of my previous concerns with this resubmission. If the authors decide to keep table 1, as requested by reviewer 3, can they specify why they chose not to include the other modifications detected by the tools that I highlighted in the first review? Nanocompore was tested to detect inosine, m5C, pseudouridine, m62A, m1G, 2'OMeA using synthetic oligos. ELIGOS was also tested on m1A, m5C, hm5C, f5C, m7G, inosine, pseudouridine, and 5moU using IVT constructs. nanoDoc was tested on "23 different types of modifications in Escherichia coli and Saccharomyces cerevisiae" rRNA.

The authors also did not address my concerns over F1 scores = 1 for methods 4 and 5 in challenge 3. This strongly suggests that tools should be tested on complex biological datasets that were not used for training in any way for future challenges.

As a suggestion for the authors if they intend to run future RMaP challenges is to ask the challengers to submit a container and execution instructions that can be easily added to a nextflow pipeline, like MasterOfPores or NanOlympicsMod for the authors to run on a separate dataset not presented to the challengers as a way to address the tools performance when not being run by the developers of the tools. This could simulate the user experience for these tools and allow the authors to update the challenge as needed.

Reviewer #3

(Remarks to the Author)

I checked the rebuttal letter and the revised manuscript. Authors have addressed my concerns.

Version 2:

Reviewer comments:

Reviewer #2

(Remarks to the Author)

The authors have addressed all my comments.

Predicting RNA modifications by nanopore sequencing: The RMaP challenge

Reviewer comment

Reviewer #1

- 1) Eighty percent of the dataset was used for method training, with predictions based on the remaining 20%. The predictive accuracy of these methods should also be evaluated on independent IVT datasets, such as Curlcake, ELIGOS, or IVET. Additionally, the top-performing methods should be tested in real-case scenarios using in vivo DRS datasets.

We want to thank you reviewer #1 for the interesting question. We discussed this question intensively and reached a common opinion. Even if in principle exciting, we fear that such an “a posteriori” evaluation– long after the challenge is over – is neither feasible nor appropriate, particularly for reasons of fairness to the participants. We also fear that adding this new „additional challenge“ to the manuscript will not bring value or additional information in it, but rather make it redundant and confusing in the context of a “challenge competition”. We would like to explain the reasons in more detail below:

- I. The RMaP challenge was designed in such a way that clearly defined rules and tasks were made available to all participants at the same time at the start of the challenge. Participants then had to complete these tasks within a strictly defined time frame. Only during this slot, they had access to the data, were able to work on the requests and submit their solutions within this strict time frame (also this part is now better explained in the introduction. See lines 116-131). No more solutions were accepted after this deadline. This has given everyone involved planning security and a fair and uniform set of conditions. Until the challenge was launched, no one had even knowledge of the type of data or modification (except for the labs that had designed, produced or hosted the data. However, these labs were automatically excluded from participating).

This circumstance no longer applies, and some labs may have significantly optimized their own algorithms in the meantime and additionally trained them also with the challenge data – now knowing where to expect which modification. This means that some algorithms now have a significant competitive advantage - even if this only consists of the fact that they have “learned” various features based on the challenge data. Therefore, we think that equal conditions and opportunities are no longer given for all groups.

- II. We don't see much added value of benchmarking methods that were designed and optimized by the challenge participants for a specific task, namely, to optimally answer the "challenge tasks", on a completely different data set under completely different conditions. Adding a new IVT datasets to the challenge means adding another sub-challenge inside each of three challenges. Furthermore, the quantification of false positives is already included in the three challenges themselves, since the target frequencies for the modification were created by combining modified and IVT reads to generate expected values for each modified position. This, in our view, will only create repetition within the logic of the competition.

- III. Solutions in line of our challenge were quantified by the challenge designer to evaluate the different metrics used in the manuscript in order to compare pipelines and extract key points for future development on modifications detections. In doing so, we were able to rely in our data, since we generated the data ourselves using the latest standards in nanopore technology and bioinformatic preprocessing of the data provided. We knew exactly how many modifications were placed and at which position. However, the suggested data sets (Curlicake and ELIGOS) are data sets that was created several years ago, which - albeit undoubtedly pioneering work at the time - on the one hand with technology that was still unstable at the time and with (by today's standards) outdated software and bascalling algorithms. It is certainly immensely interesting for validation, when a new method is presented to the community and is supposed to prove its performance on different data sets. However, in the setting of our challenge, we cannot see advantage in benchmarking the methods that have already competed against each other on our data against each other on another dataset.

We hope that Reviewer #1 is in line with our point of view after suggesting, in our opinion, that adding additional IVT data long after the challenge is already over, is not in line with the purpose of the RMaP challenge itself. Additionally, it wouldn't be fair to the participants to change the challenge-conditions and tasks retrospectively. We even see it as highly problematic to 'force' all participants who have followed all the rules and have already fulfilled all the obligations within the framework of the challenge and the tasks defined there to additionally deal intensively with a further validation that was never planned and documented in the challenge agreements.

However, we see it as a very good suggestion for the second round of the RMaP Challenge which is planned for summer 2025. Here, we will implement the proposals and thank you in advance for the suggestion.

2) All the DRS data is missing, which should be publicly available.

We introduced a “data availability” section, where information on how to find the public available RMaP challenges DRS datasets are provided.

- 3) The reference sequences as templates for in vitro transcription should be provided.

We agree with reviewer #1 and we added a supplementary data file called “Supplementary_data_DRS_IVT_sequence” that shows the in vitro transcription (IVT) sequence for the three challenges.

- 4) The methods of in vitro transcription, DRS library construction and ONT sequencing should be provided.

Methods section now shows four new sections called “IVT design”, “Data Generation”, “Determining Modification Target Rates” and “Data Preparation Commands” that explain how the data were generated, created and designed for the three challenges. These sections were added to the beginning of the method section.

- 5) The github page https://github.com/novoalab/RMaP_challenge is not available.

The repository was private. Now it is public and the GitHub page is now available.

- 6) In figure 1C, Challenge 2, Whether is Genome seq or RNA seq?

We apologize for the miss labelling. In Figure 1C Challenge 2 it should be RNA seq. Now it shows correctly the label “RNA seq.”

Reviewer #2

- 1) Although the details on the challenge data are described in the results section, there is no mention on how they were generated in the methods section. Details on the IVT reactions, library preparation, sequencing conditions, and software versions would help in evaluating the conclusions drawn from these challenge data and replicating the results from others in the community.

We apologise for this important part being missing so far. As mentioned before (see criticism from Reviewer Reviewer #1), the Methods Section now shows four new parts called “IVT design”, “Data Generation”, “Determining Modification Target Rates” and “Data Preparation Commands” that explain how the data were

generated, created and designed for the three challenges. These sections were added to the beginning of the method section. We also explain the sample composition in terms of frequency.

- 2) “Furthermore, details regarding how the ground truth of the modification ratios were established would help understand how the tools performed. Were the ratios of modified to unmodified established using the same basecaller and alignment versions as the submissions, or were they determined by one set of basecaller and alignment conditions?”

The method section now presents the section “Determining Modification Target Rates” where we explain how the ground truth of the modification ratios was established. Additionally, prompt line for the data preprocessing for data generation are shown in the section “Data preparation Commands”. To generate the RMaP challenge data sets, we used the same command lines for the three challenges to keep consistency in the data generation. With the additional methods sub-sections we think that now it is more clear (see line 401-405).

- 3) “I think that the discussion could spend more time highlighting how challenging data generation is for these types of challenges and that there is a need for highly characterized samples, akin to genomes in a bottle, but for RNA modifications.”

We appreciate Referee #2's important suggestion! Following the recommendation, we introduced a new paragraph in the discussion that underlines the aspect of data generation for new challenges for biological questions. The text can be found in the discussion (line 322-332).

- 4) “Given that most of the RNA modification detection tools are designed to detect m6A, I was surprised that only CHEUI made a submission for challenge2. Do the authors have any insights as to why there were so few submissions for this challenge and why the results in this challenge differ so substantially from the results seen in the paper describing CHEUI's development and deployment?”

CHEUI was used in the method 2 and method 3 submissions, which refer challenge 1 and 2 respectively. This to clarify that CHEUI was used for both m5c and m6A challenges.

Regarding the amount of submissions: In advance of the challenge, a great many participants were interested in the challenge and also made a clear statement of intent that they wanted to take part in the challenge. Based on the download and access rates, we also assume that well over 20 labs ultimately downloaded the data and worked on it. However, most labs did not submit a solution in the end, which is why we could only include the number of participants who uploaded a

solution in our manuscript. Unfortunately, we cannot say in detail how many participants actually worked on the data in the end and for what reasons no solution was ultimately uploaded, but we have summarized some of the reasons that emerged from the feedback from the participants who provided feedback:

- 1) The challenge was designed to be open for a certain period, which was exactly 45 days. After that, no additional submissions were accepted. In all likelihood, the time frame was too short and also fell during a time when many people were still on their summer holidays. Even the participants who submitted their work “on time” reported that dealing with the data was a very intensive and more time-consuming task than expected. Furthermore, several participants would have appreciated more time to optimise their results. Further feedback from participants who had only worked on individual sub-challenges indicated that they would have worked on at least another sub-challenges if they had more time. However, the given framework only allowed focusing on one of the sub-challenges.
- 2) Regarding the difference in performance of CHEUI between the two drafts, we can’t say why it happen. Additional work is needed to investigate this difference, which goes beyond the RMaP objectives. Hypothetically, it could be that the dataset create here for the RMaP challenges are “special cases” that CHEUI may not have been well trained for. As mentioned from Reviewer #2, data generation for neural networks training can be challenging. Thinking in this direction, we are maybe in the few percentages special case that CHEUI cannot handle, despite the fact the CHEUI can performs with high accuracy values in several other scenarios. However, this is just a hypothesis and to judge that, a study (outside the RMaP Challenge) would be necessary, which systematically examines the performance differences of CHEUI on different data sets and with different user settings.

- 5) I think table 2 would be more useful to appear in the document before figure 2

We move the Table 2 before Figure 1. In this way the full results can be observed to the beginning of the results sections.

- 6) Table 1 doesn't really seem necessary for the manuscript.

We see the point of Reviewer #2, but we would prefer to keep the table because can help readers to have a starting list to look up for their future work. Additionally Reviewer #3 explicitly even proposed an extended version of this table.

Reviewer #3

- 1) “The main purpose of RMaP challenge is not so clear to me. To compare different pipelines using current algorithms? To develop new state-of-art algorithms?”

As suggested by Reviewer #3, we also agree that the main purpose of the RMaP challenge should be discussed more in details in the introduction. In this regard, we changed the last part of the introduction and RMaP challenge purpose explained in more detail.

- 2) “Authors linked “the higher performance of Method 1 on frequency detection” with “the accuracy of basecalling from Dorado”. Which is less interested since the focus of challenges should be the modification prediction. Which also makes the comparison between “Bayesore” and “CHEUI” not informative. Authors also suggested “that the post-basecalling and alignment pipeline significantly impacts the analysis.” (Line 261) Which also makes the comparison uninformative.

A controversial statement appeared in Line 264, “suggests that the basecaller has a minor impact on the analysis.”, we do not know. Meanwhile, different versions of Dorado (v0.3.4 & v0.3.2), Guppy (v6.0.6, v6.4.2 & v6.5.7) & minimap2 (v2.26 & v2.26) were employed during analysis. With or without documented parameters. Check the citations of minimap2.”

The comment of Reviewer #3 is taking two topics:

- 1) comparison between two methods in challenge 1
- 2) the statement on line 264, which doesn't consider versions of the basecall platform.

Answer to the comment (1):

One of the goals of the challenge is also to compare individually developed pipelines to obtain information for further work in software development in the field of RNA-Modification detection. Following this direction, the comments and comparison between two methods in challenge 1 can support future work and networking activities. We did it by comparing pipeline steps and results reported in table 2 to find possible key points or steps that have a major or minor impact on RNA modifications detection. However, we also know that this are general or suggested guidelines and new analysis has to be performed in this direction to confirm certain hypotheses. But the reader can have important information for future pipeline development, like role of post processing or selection of basecaller. However, we think this comment can be linked with a previous one from Reviewer #3, which was asking to explain better the goal of the RMaP challenge. In this regard, we decided to change part of the final part of the introduction to explain better the RMaP challenge purpose.

Answer to the comment (2):

We see the point of Reviewer #3 on the statement of line 264 (now 268). We have modified the sentence to make the message we want to send to the reader clearer. We still think that it is possible to give some information about the basecaller. This result comes from a triangulation analysis between method 4,5 and 6. Method 5 and 6 have practically the same pipeline, except for “label pre-processing”. This suggests already that the second step after basecalling is more important (under certain limitation of accuracy for the basecaller). Supporting this idea, is the fact that method 4 and 5 have both high performance but different basecaller, which indicates again that basecalling has a minor impact on the analysis when the basecaller has enough accuracy in nucleotide detection.

About the model versions, it is not really clear from ONT sites, but we investigate there and it is possible to see that basecaller model and software model (like Dorado) are not linked to each other. In fact, basecaller models have their own version number and they can be called between different software versions, unless it is not specified in the description of the model (for example in some RNA modification calling model). For normal basecalling, it is not the case.

- 3) “Table 1 is not sufficient since many algorithms are missing here. Publication information or download links should be included.”

We thank the Reviewer for the recommendation. We extended the list of methods to follow the request of Reviewer #3 and added a column that shows the reference number as suggested. However, we would like to mention that this table was not intended to give a complete listing of all algorithms ever developed by the community for the detection of RNA modifications using direct RNA sequencing. Such a systematic and comprehensive review would go far beyond the scope of this manuscript, which has a completely different focus.

Here we rather wanted to provide a short tabular list of the methods mentioned in the introduction. Nevertheless, following the Reviewer’s recommendation, we have included further important methods, especially those that have been developed recently. If we have nevertheless overlooked important methods that should be considered here, we would be very grateful for targeted suggestions from the reviewer.

- 4) Line 172, Line 210 & Line 246, authors wrote “The target frequencies range from 0.12 to 0.33.”, “The target frequencies range from 0.01 to 0.1.” & “The target frequencies range from 0.31 to 0.5.” Did authors tune the target frequencies by adjust the mix ratio of modification data and plain data?

We added in the methods a sub-section that explains how the Challenge RMaP data sets were created (called “Determining Modification Target Rates”). Here, we

explained how the expected modification frequencies were created, which is a mixture between unmodified and modified data sets.

- 5) Fig1. Affiliation/Institute contributions are missing. Author contributions are missing.

Thank you for the suggestion. We didn't notice it and it is an important element to check for not leaving anyone of the participants left apart. Indeed comparing fig. 1 and the list of participants, there is a miss match of 2 elements between them. We investigate and double checked the affiliations/Institute in Fig. 1 with the list of authors and we observed that 2 groups have the same logo, which creates this miss match and a bit of confusion.

Regarding the contributions, we have added the section 'Author contributions' at the end of the manuscript.

- 6) Line 219, authors wrote "which is caused by a large number of false positive predictions." I guess not only false positive, but also false negative.

We agree with Reviewer #3. Now the text is change as following:

"While the accuracy is slightly worse for Method 3, the F1-score is way lower, which can be caused by a large number of false-positive predictions, false-negative predictions, or a combination of both." (line 224-226)

- 7) Line 233, "an artificial DNA sequence", is it a random sequence or a thoughtful sequence for purpose?

To answer this question, explanation of synthetic data generation was added to the beginning of the Methods section ("IVT design" and "Data Generation", line 350 and 357 respectively). In details, the sequence was designed to include all the possible 5-mer combinations for a modified 5-mer having the RNA modified bases in the centre.

Predicting RNA modifications by nanopore sequencing: The RMaP challenge

Reviewer comment

Reviewer #2

- 1) The authors addressed most of my previous concerns with this resubmission. If the authors decide to keep table 1, as requested by reviewer 3, can they specify why they chose not to include the other modifications detected by the tools that I highlighted in the first review? Nanocompore was tested to detect inosine, m5C, pseudouridine, m62A, m1G, 2'OMeA using synthetic oligos. ELIGOS was also tested on m1A, m5C, hm5C, f5C, m7G, inosine, pseudouridine, and 5moU using IVT constructs. nanoDoc was tested on "23 different types of modifications in Escherichia coli and Saccharomyces cerevisiae" rRNA.

We apologize to reviewer#2 for the miss-understanding and thank you for the important hint. Even though Nanocompore and EliGOS were mentionend in the text, they were mislabeled and not included in the table by mistake. Now they are correctly labeled in Table 1 as well as in the main text. We also introduced NanoDoc in the list that was indeed completely missing. Thank you again for the suggestions to make the list more accurate and appropriate.

- 2) The authors also did not address my concerns over F1 scores = 1 for methods 4 and 5 in challenge 3. This strongly suggests that tools should be tested on complex biological datasets that were not used for training in any way for future challenges. We agree with reviewer #2 that this should indeed be formulated more precisely. To emphasize the aspect of the F1 score for method 4 and method 5 and the need to test the algorithm on complex biological datasets, we have therefore added a small paragraph in the discussion section. Here, we explain that this aspect is evidence that the algorithm obviously works well for a specific problem, but that it probably can not be generalized to the analysis of real biological systems. (line 322). We hope that this additional clarification has now addressed the concerns of reviewer #2 regarding our wording.

- 3) As a suggestion for the authors if they intend to run future RMaP challenges is to ask the challengers to submit a container and execution instructions that can be easily added to a nextflow pipeline, like MasterOfPores or NanOlympicsMod for the authors to run on a separate dataset not presented to the challengers as a way to address the tools performance when not being run by the developers of the tools. This could simulate the user experience for these tools and allow the authors to update the challenge as needed.

We want to thank you Reviewer#2 for the suggestion. We like the idea of adding a pipeline to know platform as can be nextflow or Epi2ME, which can help to run the algorithm for problems not linked to the challenge. This was the first RMaP challenge and we appreciate any feedback that can improve future publications in this direction to have an easier access to the published work and help the reader to re-use pipelines for their goal.